# Phenome-wide analysis of Taiwan Biobank reveals novel glycemia-related loci and genetic risks for diabetes

Chia-Jung Lee[1,13,14], Ting-Huei Chen[2,3,14], Aylwin Ming Wee Lim [1,4], Chien-Ching Chang[1], Jia-Jyun Sie[5], Pei-Lung Chen[6,7,8], Su-Wei Chang[9,10], Shang-Jung Wu[1], Chia-Lin Hsu[1], Ai-Ru Hsieh[11✉], Wei-Shiung Yang[6,8,12✉] & Cathy S. J. Fann[1✉]

To explore the complex genetic architecture of common diseases and traits, we conducted comprehensive PheWAS of ten diseases and 34 quantitative traits in the community-based Taiwan Biobank (TWB). We identified 995 significantly associated loci with 135 novel loci specific to Taiwanese population. Further analyses highlighted the genetic pleiotropy of loci related to complex disease and associated quantitative traits. Extensive analysis on glycaemic phenotypes (T2D, fasting glucose and $HbA_{1c}$) was performed and identified 115 significant loci with four novel genetic variants (*HACL1*, *RAD21*, *ASH1L* and *GAK*). Transcriptomics data also strengthen the relevancy of the findings to metabolic disorders, thus contributing to better understanding of pathogenesis. In addition, genetic risk scores are constructed and validated for absolute risks prediction of T2D in Taiwanese population. In conclusion, our data-driven approach without a priori hypothesis is useful for novel gene discovery and validation on top of disease risk prediction for unique non-European population.

[1] Institute of Biomedical Sciences, Academia Sinica, Taipei 115, Taiwan. [2] Department of Mathematics and Statistics, Laval University, Quebec, QC G1V0A6, Canada. [3] Brain Research Centre (CERVO), Quebec, QC G1V0A6, Canada. [4] Taiwan International Graduate Program in Molecular Medicine, National Yang Ming Chiao Tung University and Academia Sinica, Taipei 115, Taiwan. [5] Department of Mathematics, National Changhua University of Education, Changhua, Taiwan. [6] Graduate Institute of Medical Genomics and Proteomics, College of Medicine, National Taiwan University, Taipei 10617, Taiwan. [7] Department of Medical Genetics, National Taiwan University Hospital, Taipei 100225, Taiwan. [8] Graduate Institute of Clinical Medicine, College of Medicine, National Taiwan University, Taipei 10617, Taiwan. [9] Clinical Informatics and Medical Statistics Research Center, Chang Gung University, Taoyuan 333, Taiwan. [10] Department of Laboratory Medicine, Chang Gung Memorial Hospital at Linkou, Taoyuan 333, Taiwan. [11] Department of Statistics, Tamkang University, New Taipei City 251301, Taiwan. [12] Department of Internal Medicine, National Taiwan University Hospital, Taipei 100225, Taiwan. [13] Present address: Department of Pathology and Immunology, Washington University School of Medicine, Saint Louis, MO, USA. [14] These authors contributed equally: Chia-Jung Lee, Ting-Huei Chen. ✉email: airudropbox@gmail.com; wsyang@ntu.edu.tw; csjfann@ibms.sinica.edu.tw

Genetic epidemiological methodologies such as genome-wide association studies (GWAS), phenome-wide association study (PheWAS), conditional GWAS, genetic correlation[1], and Mendelian randomization (MR)[2] have allowed elucidation into the convoluted interplay between genetics and phenotypes of complex diseases in human. Conventionally, genetic epidemiological studies focused on specific phenotypic or disease trait, which, to date have reported more than 370,000 associations over 1800 traits[3]. However, complex diseases require more in-depth parallel analysis due to the heterogeneity and genetic pleiotropy of complex diseases. The abundance of both genotype and phenotype data from biobanks such as the UK Biobank (UKBB)[4] and the Biobank Japan (BBJ)[5] has allowed extensive phenome-wide genome-wide association studies of quantitative traits and diseases. However, most of the previous discovery are based on subjects with European ancestry and gradually, the establishment and maturation of biobanks with non-European ancestry have revealed the importance of diversity in genetic epidemiology[6].

The population-based biobank of Taiwan (Taiwan Biobank, TWB) was launched in 2012 and encompasses 0.61% of the total population of Taiwan with more than 144,000 participants, which is comparable to the population coverage of BBJ (0.16%) and UKBB (0.74%). The majority of Taiwanese (over 99%) are of Han Chinese ancestry who migrated from mainland China[7,8]. Analyses based on phenotypes in TWB may reveal significant genetic effects of critical health-related traits in Taiwanese and in extension, other populations with East Asian ancestry.

In this paper, we reported a comprehensive PheWAS of 10 diseases and 34 quantitative traits from TWB with complementary conditional analysis, genetic correlation and MR. As most of the available traits are cardio-metabolically related, we focused our analysis on type 2 diabetes (T2D). T2D has been one of the leading causes of mortality and morbidity in Taiwan with high socioeconomic burden[9]. The T2D prevalence in Taiwanese population was estimated to be 11.6% in 2016[9] with the average age of onset around 59.5 years old[10]. In addition, the mean annual cost for a patient with T2D-related major complications was estimated to be USD $4189[11]. T2D is a heritable trait compounded by various degrees of gene–gene and gene–environment interactions with heritability estimates ranging from 20 to 80%[12]. The polygenic risk score (PRS), which aggregates the effects of multiple disease-associated genetic variants, has previously been used for T2D risk prediction in various populations. To further improve the applicability of PRS on identifying high-risk individuals for early intervention specifically for Taiwanese, we constructed absolute risk models which incorporated various risk factors for estimation on the probability that an individual free of T2D at a given age will develop T2D in an upcoming time interval.

Our findings of novel candidate genes (HACL1, RAD21, ASH1L, and GAK) related to T2D in Asian population provided insights into the pathophysiology of T2D, and could be potential targets for clinical diagnosis and therapeutic interventions.

## Results

### Phenome-wide association analysis for 10 binary and 34 quantitative traits in TWB. 
Supplementary Data 1 summarizes the PheWAS results for ten binary and 34 quantitative traits in TWB. Detailed demographic data are presented in Supplementary Data 2. About 6 million imputed autosomal SNPs passed our QC criteria and were tested for associations with each of the 44 traits. The 44 traits were grouped into nine categories (Supplementary Data 1): anthropometric ($n = 5$), metabolic ($n = 8$), cardiovascular ($n = 6$), hematological ($n = 5$), kidney-related ($n = 6$), liver-related ($n = 6$), stress-related ($n = 5$), pulmonary-immunological ($n = 2$), and articular-skeletal ($n = 1$).

In total, 995 significantly associated loci ($P \le 5 \times 10^{-8}$) with more than 100 loci were found to be specific to the TWB population (Supplementary Data 1 and Supplementary Data 3). TWB PheWAS results is summarized in the Fuji plot[5] (Fig. 1) with each layer corresponding to a phenotype, and only significant loci were plotted. We identified several pleiotropic regions associated with multiple phenotypic categories (Supplementary Data 4). For instance, chromosome 2 contains a highly pleiotropic region (positions 27,508,345 through 27,527,678), associated with 13 traits in four categories (hematological, kidney-related, liver-related, and metabolic). This genomic region contains a T2D-associated glucokinase regulator (GCKR). Another highly pleiotropic region on chromosome 12 (positions 111,792,215 through 112,136,812) was associated with eight traits in five different categories (anthropometric, kidney-related, liver-related, vascular-metabolic, metabolic, and FVC). This pleiotropic region on chromosome 12 contains two genes (ALDH2, TRAFD1); genetic variants of ALDH2 have been shown to be associated with risk to T2D, micro-vascular and macro-vascular complications[13,14]. Further details of the PheWAS results are available in Supplementary Data 4.

To control for the confounding effect of LD, conditional association analyses were carried out with adjustment for corresponding lead SNPs. We found further 115 independent significant signals (lead SNP $r^2 \le 0.1$; $P \le 5 \times 10^{-8}$). Of these signals, 38 were mapped to genes different from their corresponding lead SNPs (Supplementary Data 5). Of the 15 TWB unique signals (Supplementary Data 5), 6 were mapped to genes different from their lead SNP in conditional analyses. This result demonstrated the power of conditional analysis to resolve confounding effects due to LD within an associated region, and to discover putative candidate genes that might be missed by marginal association analyses.

### Shared genetic architectures between traits. 
To elucidate the underlying mechanisms of identified associations, we estimated genetic correlations between each pair of quantitative traits and binary diseases using bivariate LD score regression[1] as shown in Fig. 2, Supplementary Fig. 1, Supplementary Data 6, and Supplementary Data 7. Of the 101 genetic associations identified with FDR $\le 0.05$, the strongest signal was found between hypertension and mean arterial pressure ($r_g = 0.846$, FDR $= 4.93 \times 10^{-102}$). For the vascular-metabolic traits, T2D was found to be significantly associated with 27 quantitative traits, such as $HbA_{1c}$, BMI, WC (Supplementary Data 6 and Supplementary Data 7). As expected, $HbA_{1c}$ has the strongest correlation ($r_g = 0.756$, FDR $= 1.06 \times 10^{-22}$) with T2D. Moreover, we identified several association signals previously reported by BBJ[5] as well as other studies. For example, we observed significant correlations between T2D and several kidney-related traits, such as microalbumin[15] ($r_g = 0.661$, FDR $= 1.90 \times 10^{-3}$), uric acid[16] ($r_g = 0.260$, FDR $= 9.61 \times 10^{-5}$), and BUN[17] ($r_g = 0.171$, FDR $= 1.85 \times 10^{-2}$).

### Mendelian randomization. 
MR analyses were carried out for BMI, Waist circumference, Body fat percentage, Waist-hip ratio, Hip circumference, triglyceride, HDL-C, and VLDL-C against glycemia-related traits (T2D, $HbA_{1c}$, and FG) (Supplementary Data 9–11). We identified 180, 186, and 185 instrumental variables (IVs) at a genome-wide significance level and clumping threshold of $r^2 = 0.01$ associated with T2D, $HbA_{1c}$, and FG, respectively. Of these 551 IVs, none of them were found to have horizontal pleiotropy effects on glycemia-related traits by MR-Egger. All of

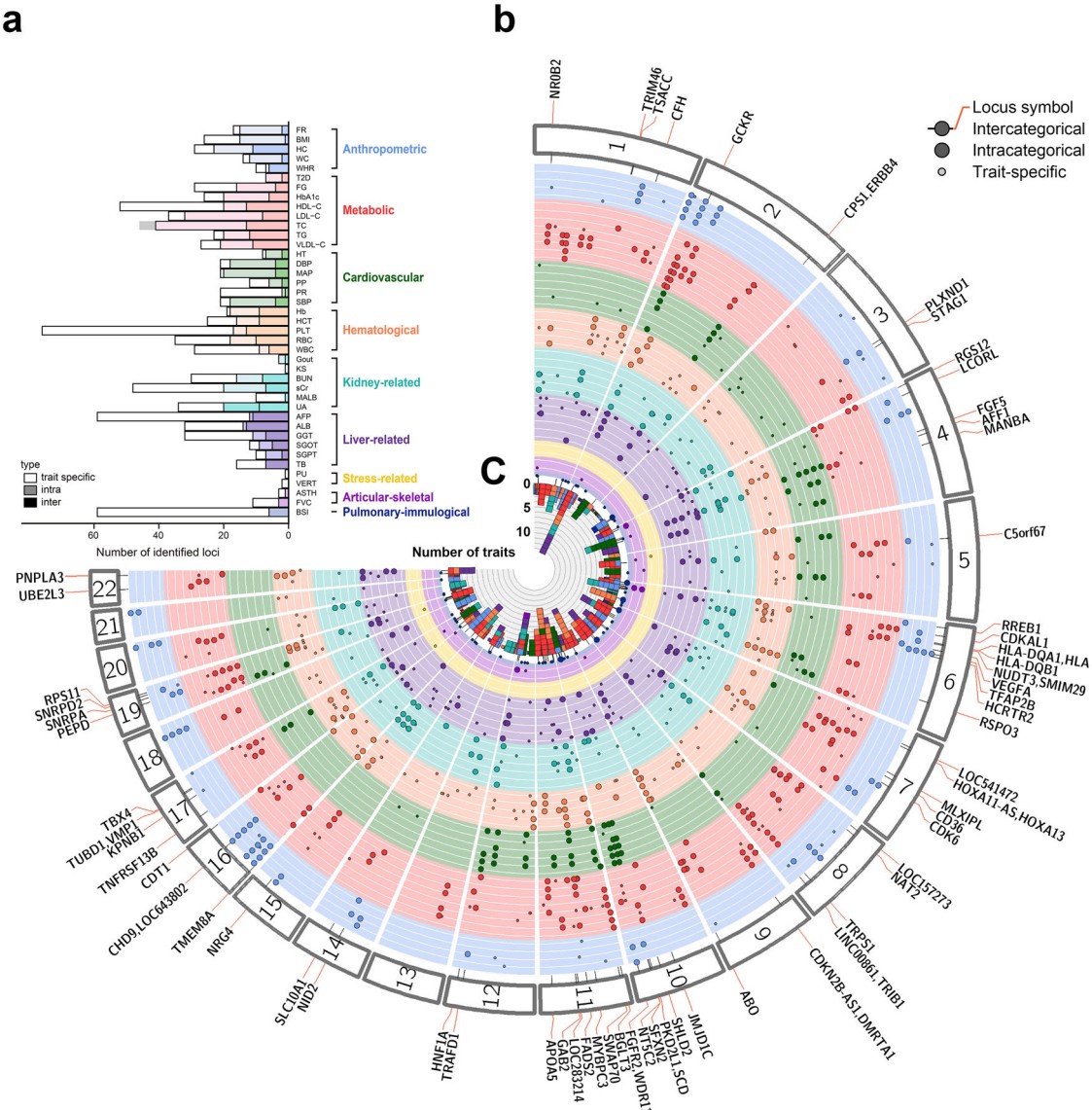

**Fig. 1 Overview of loci identified in this PheWAS and their pleiotropy. a** Number of identified loci for each trait group by trait categories. Color saturation indicates whether the locus has pleiotropic effects. White: trait-specific locus; medium saturation: shared locus within a single trait category; full saturation: shared locus between trait categories. **b** Fuji plot of the 41 traits having association signals. Each association lead SNP is presented as a dot and arranged by its physical position along the angle starting from the 12 o'clock position. Each line corresponds to a trait indicated in (**a**) and each lane is colored by the color of trait category. The larger dots indicate pleiotropic association loci. **c** The number of associated traits for each inter-categorical pleiotropic locus.

the 551 IVs have passed the SNP outlier test (unknown pleiotropic SNPs) through MR-PRESSO.

In the two-sample MR analysis on T2D, the overall causal estimate (IVW odds ratio (OR) estimate) for T2D per unit increase in BMI was 1.2566 ($P = 0.0011$), for the effect of a 1-unit increase in HDL-C on the risk of T2D was 0.9762 ($P = 0.0087$), and for the effect of a 1-unit increase in VLDL-C on the risk of T2D was 0.9832 ($P = 0.01$) (Supplementary Data 9). In the two-sample MR analysis on $HbA_{1c}$, the overall causal estimate for $HbA_{1c}$ per unit increase in BMI was 1.0317 ($P = 0.0071$) (Supplementary Data 10). As for the two-sample MR analysis on FG, the overall causal estimate for FG per unit increase in HDL-C was 0.9472 ($P = 0.0229$) (Supplementary Data 11). By contrast, Waist circumference, Body fat percentage, Waist-hip ratio, Hip circumference, and triglyceride were not found to be significantly associated with T2D, $HbA_{1c}$, and FG (Supplementary Data 9–11). After the Bonferroni correction ($P = 0.0021$ (0.05/24)), the causal relationship between BMI and T2D

remained from our MR analyses. In our study, lipid profiles such as HDL-C and VLDL-C were found to be superior to anthropometric measurements in predicting the risk of glycemia-related traits.

**GWAS of T2D and glycemia-related phenotypes.** GWAS of FG ($N = 75,627$) (Supplementary Fig. 2) identified 29 significantly associated loci (such as *CTBP1-DT, STEAP2-AS1, NOM1/MNX1,* and *GAD2*) (Supplementary Data 1), while GWAS of $HbA_{1c}$ ($N = 76,171$) (Supplementary Fig. 2) found 26 significantly associated loci including *HACL1* (Fig. 5), *CTBP1-DT,* and *C5orf67*. GWAS of 63,177 non-diabetic controls (94.3%) and 3844 T2D subjects (5.7%) (Supplementary Fig. 2) revealed seven SNPs significantly associated with T2D (Supplementary Data 1, Supplementary Data 3): *CDKAL1, MIR129-1, LEP, SLC30A8, MED30, CDKN2B-AS1, DMRTA1, CDC123, CAMK1D, KIF11, HHEX,* and *KCNQ1*. Collectively, we identified 41 genes significantly associated with T2D and glycemia-related phenotypes

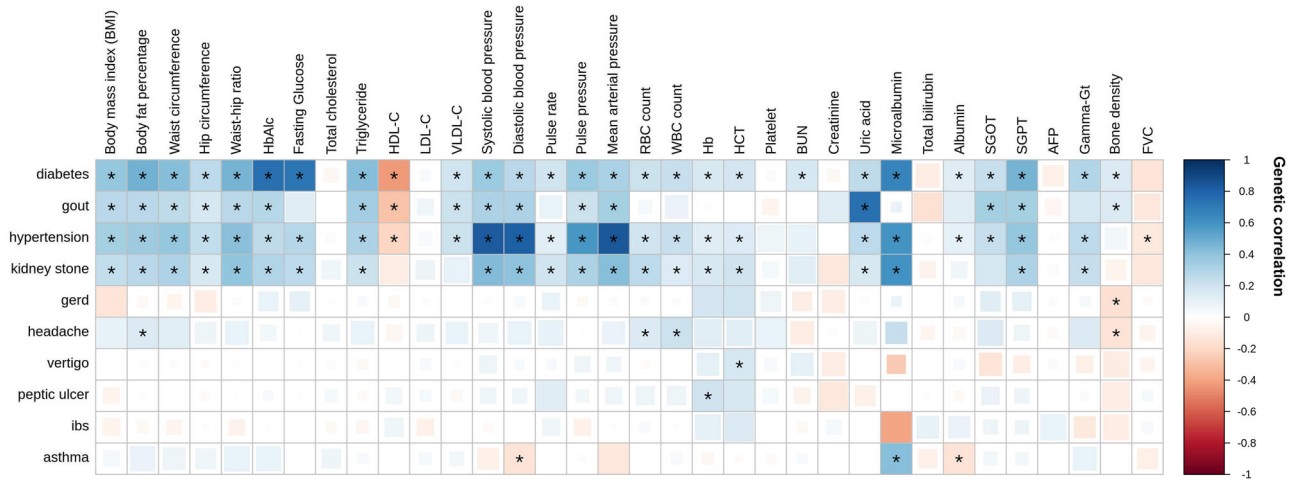

**Fig. 2 Genetic correlation between binary and quantitative traits.** Pairwise genetic correlations ($n = 946$) were estimated by bivariate LD score regression (full correlation results are shown in Supplementary Fig. 2). Only correlations between 10 binary traits and 34 quantitative traits ($n = 340$) are presented in this figure. Positive and negative correlations are colored in blue and red, respectively. The intensity of correlation is indicated by the color saturation. The FDR is calculated by the Benjamini–Hochberg method. Size of the color block represents the FDR of each correlation, and significant correlations (FDR ≤ 0.05) are indicated by asterisks.

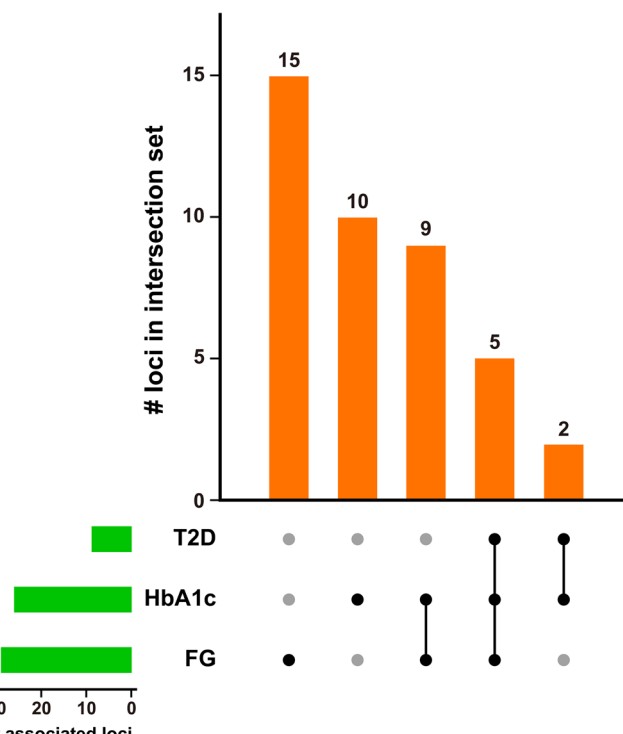

**Fig. 3 Overlapping loci within glycemic traits.** The upset plot[59] summarizes shared loci of three glycemic traits. The dot-and-line chart in the bottom combination matrix indicates intersections between three traits. For example, the second column from the left indicates a set of loci only associated with HbA1c, while the middle column indicates loci associated with both HbA1c and FG. The upper bar chart shows the number of associated loci in each set. The lower left horizontal bar chart represents the number of loci associated with T2D (7), HbA1c (26), and FG (29).

with some SNP to be shared among the glycemic traits and T2D (for e.g.,: MED30/SLC30A8, CDC123) (Supplementary Data 8 and Fig. 3). Summary of SNPs and mapped gene by FUMA SNP2GENE are shown in Supplementary Data 12–14, Supplementary Table 1, and Supplementary Figs. 3–4. From the SNPs

identified, most of them are located in intronic and intergenic region; only a minuscule number of SNPs are located in the exonic region (T2D: 0.3%, HbA1c: 1.4%, Fasting glucose: 0.8%) (Supplementary Table 1 and Supplementary Fig. 3). The mapped genes were not statistical significantly expressed in any specific tissue types (Supplementary Fig. 4). However, we noticed upregulation of differentially expressed genes in glycaemic-related tissue types such as kidney, pancreas, stomach and adipose tissues. A summary of the functional annotation of the mapped genes are available in Supplementary Data 12–14.

**Conditional association analyses.** Conditional association analysis using the lead SNPs as covariates was performed for all SNPs in each locus. Ten SNPs were significantly associated with FG independent of their lead SNPs. Of these, eight were associated with the same genes as their lead SNPs despite low $r^2$; while the other two were located on different genes (Supplementary Data 5). For instance, rs742763 and its lead SNP rs9380826 were all mapped to *GLP1R*. In contrast, rs2632372 and its lead SNP rs1402837 were respectively mapped to *NOSTRIN* and *G6PC2*. Fourteen SNPs were associated with HbA1c independent of their lead SNPs with eight SNPs mapped to the same genes as their respective lead SNPs (Supplementary Data 5). For example, an independent SNP rs75151020 and its lead SNP rs742761 were both mapped to *GLP1R*. In contrast, rs1326821916 and its lead SNP rs72501962 were separately mapped to *GAK* and *CTBP1-DT*.

For T2D, the associations of two SNPs (rs11994747 and rs115894051) on different loci remained loci-wide significant ($P \leq 1 \times 10^{-5}$) after adjustment. (Supplementary Data 5). In contrast, rs11994747 was associated with T2D independent of its lead SNP rs35859536. The potential genes for lead SNP rs35859536 are *SLC30A8/MED30*, whereas rs11994747 is located in the intronic region of *RAD21* (Supplementary Data 5). *RAD21* has never been reported to be associated with T2D or any glycemia-related phenotypes.

The conditional association analyses revealed another 26 additional independent loci (such as *RAD21, CAMKMT/ LINC01833, NOSTRIN, ASH1L, GAK, POLD2/MYL7, SND1, STARD13,* and *LUC7L*). NOSTRIN, POLD2/MYL7, CAMKMT/ LINC01833, and *SND1* were previously reported to associate with glycemia-related phenotypes[18–21]. *STARD13* and *LUC7L* were

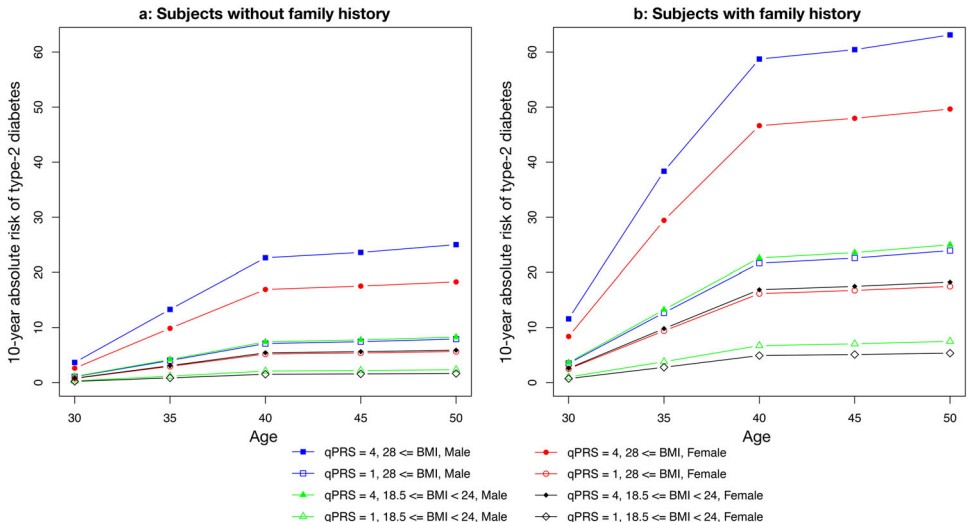

**Fig. 4 10-year absolute risk of type 2 diabetes for subjects without and with a family history of type 2 diabetes by age and risk profiles. a** Subjects without a family history. **b** Subjects with family history.

associated with hemoglobin[22] and thus not considered to be associated with glycemic traits.

**Absolute risk of developing T2D in the Taiwanese population.** The absolute risk modeling for T2D was based on variables including PRS quintile, family history of T2D, BMI, and sex. The 10-year absolute T2D risk demonstrated a significant risk separation across different combinations of PRS, BMI and sex in the Taiwanese population aged 30 to 50 with or without the presence of family history of T2D (Fig. 4 and Supplementary Data 15). For instance, a 40-year-old male without T2D family history, with BMI > 28 and qPRS = 4 (the 4th PRS quintile), has an estimated 10-year absolute T2D risk of 22.6% compared to 7.0% for a male of the same age and qPRS but with normal BMI, and to 7.1% for a male of the same age and BMI but lowest qPRS. Through our analysis, BMI was identified to have the highest risk, which can increase the probability by 15.6% compared to normal BMI, when given other risk values were the same. PRS also gave a similar effect with 15.5%. Validation results based on prospective cohort samples showed that the model had good calibration of relative risk for all sub-groups except for the sub-group of males aged older than 45 years as assessed by the chi-square goodness of fit test shown in Supplementary Fig. 5. Absolute risk (AR) had good calibration for the sub-group of male aged younger than 45 while the observed AR in TWB cohort sample were lower than the model projected ones, assessed by Hosmer-Lemeshow goodness of fit test.

## Discussion

Here we present a large-scale PheWAS of ten binary and 34 quantitative traits in 77,072 Taiwanese participants from TWB, identifying 995 association signals. This information in combination with the data from other ancestries or geographic regions will provide more insight into the genetic architecture of cardiometabolic traits for people from different parts of the world. Wei et al.[23] showed that TWB cohort represented diverse ancestry of the different province of mainland China and 99% of TWB cohort were Han Chinese. Genetic architecture of Han Chinese is relatively similar to other East Asian populations such as JPT and KHV population from 1000 genome. For diabetes, comparisons of the genetic data from different populations may help to

decipher the genetic mechanisms underlying the pathophysiology of T2D for individuals of different ancestries.

As demonstrated in the Fuji plot, we identified several highly pleiotropic loci and discovered putative shared genetic effects for several diseases or traits. For example, *GCKR* shows interesting pleiotropic effects. Previously, it has been shown to be associated with T2D[24], gestational diabetes[25], triglyceride[26], and fatty liver[24]. *GCKR* encodes a regulatory protein for glucokinase (*GCK*), regulating the subcellular localization and allosteric switch of GCK, the rate-limiting enzyme for cellular glucose uptake[27,28]. These results clearly demonstrate the strength of PheWAS in identifying genes with pleiotropism on T2D and its comorbid traits. The lead SNP rs6547692 for *GCKR* gene has a similar prevalence across TWB (MAF = 0.485), BBJ (MAF = 0.4387) and UKBB (MAF = 0.44). The regional association plot for rs6547692 (*GCKR*) is shown in Fig. 5.

Our GWAS analysis revealed novel association of *HACL1* with HbA$_{1C}$ (rs1481559294, $P = 4.42 \times 10^{-8}$) in 77,072 individuals. The recent largest trans-ancestral GWAS of glycemic traits showed marginal significance ($P < 1 \times 10^{-3}$) of several single-nucleotide variations within *HACL1* (chromosome 3 from genomic position of 15,560,699 to 15,601,569) in sub-population of around 10,000 individuals[29]. Their meta-analysis results consist of 281,416 individuals with 13% East Asian. From Genome Aggregation Database (gnomAD)[30], we noticed that rs1481559294 for *HACL1* was a relatively rare variant (African $n = 42,024$ MAF = 0.00002, East Asian $n = 3132$ MAF = 0.0006) as compared to the MAF in our population of 0.0013 ($n = 77,702$). The regional association plot for rs1481559294 (*HACL1*) is shown in Fig. 5. *HACL1* encodes for the enzyme 2-hydroxyl-CoA lyase 1 which is involved in catalyzing the conversion of even-chain fatty acids into odd-chain fatty acids by cleaving C1 in peroxisome fatty acid α-oxidation[31]. Jenkins et al. showed that Hacl1 knockout mice had lower plasma and liver C17:0 fatty acid, but did not observe significant difference in adipose tissue[32]. Gene expression of *HACL1* has low tissue specificity, however, transcriptomics data showed tissue expression of *HACL1* clustered in the intestine and liver, associated with lipid metabolism (Supplementary Fig. 6). Kocarnik et al. previously reported that SNP rs73148185, which was mapped to *HACL1*, was associated to C-reactive protein level in a multi-ethnic population[33]. Chronic systemic inflammation signified by

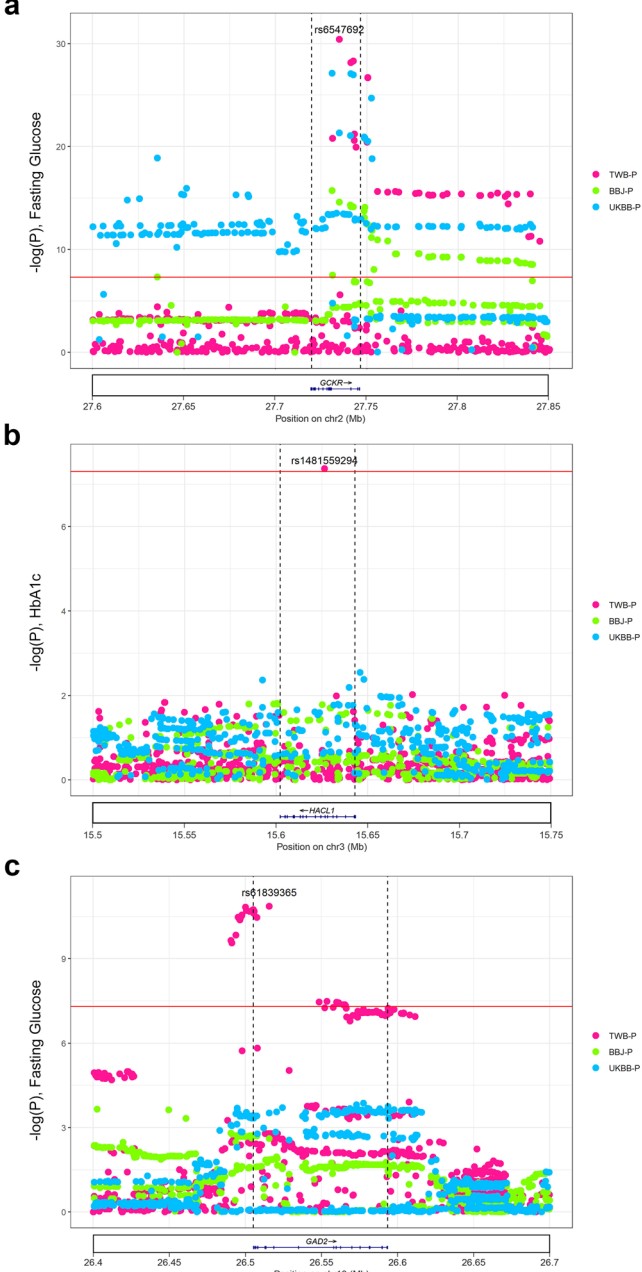

**Fig. 5 Regional association plots, identified by comparison of Taiwan Biobank (TWB), Japan Biobank (BBJ) and UK Biobank (UKBB) genome-wide association studies (GWASs).** The $X$ axis represents the position of loci (hg19). The $Y$ axis represents $-\log_{10}(P)$. Red dots are $P$ values of variants from TWB. Green dots are p values from the BBJ. Blue dots are $P$ values from UKBB. The red horizontal line represents $P = 5 \times 10^{-8}$. The black dashed line represents the location of the gene. **a** GCKR gene; **b** HACL1 gene; and **c** GAD2 gene.

elevated C-reactive protein level is a key underlying pathophysiology in patients with T2D[34]. PheWAS of rs1481559294 showed association with glycemic traits of HbA$_{1C}$ and FG while displaying nominal significant association with known cardiometabolic risk factors such as waist circumference and other anthropometric traits (Supplementary Fig. 7). The tissue expression HACL1, association with known T2D risk factors and relevancy in lipid metabolism could indicate an indirect, yet important genetic variant associated with glycaemic trait specifically in the Taiwanese population.

Another intriguing finding from our GWAS was the association of GAD2 (Glutamate decarboxylase 2) (rs61839365, $P = 1.37 \times 10^{-11}$) with fasting glucose. GAD2 (rs2839671, $P = 8.95 \times 10^{-9}$) was recently reported in the same trans-ancestral GWAS of glycaemic traits by Chen et al.[29]. Despite having a smaller sample size, we managed to identify the association of GAD2 with fasting glucose, highlighting the importance of genotype data from diverse ancestries such as Taiwan Han Chinese. The lead SNP rs61839365 for GAD2 is more prevalent in East Asian with MAF of 0.293 in TWB and 0.416 in BBJ as compared to 0.18 in UKBB. The regional association plot for rs61839365 (GAD2) is shown in Fig. 5. GAD2 is a major auto-antigen in autoimmune-associated type 1 diabetes and in a subset of T2D, latent autoimmune diabetes in adults[35]. Glutamate decarboxylase 2 catalyzes the formation of gamma-aminobutyric acid (GABA). GABA is an inhibitory neurotransmitter that is a critical component of neurophysiologic function. Upon the stimulation of glucose, GABA, co-secreted with insulin, has been shown to inhibit glucagon secretion via the activation of GABA$_A$-receptor chloride channels of α cells[36]. It has also been documented that beta cells secrete GABA in a pulsatile manner in synchrony with insulin secretion[37]. The storage and secretion of GABA in beta cells are defective in islets of type 1 and type 2 diabetic patients[37]. Taken together, it is plausible that GAD2 may modulate the blood glucose level by regulating glucagon secretion. GAD2 is highly expressed in brain tissues (especially the hypothalamus) (Supplementary Fig. 6), strengthening the understanding of key role of the hypothalamic–pituitary–adrenal axis in neuroendocrine dysregulation of T2D[38].

The conditional association analyses revealed three novel genetic association with glycaemic traits (RAD21, ASH1L and GAK). The regional association plots (before and after conditional analysis) for rs11994747 (RAD21), rs371382391 (ASHIL) and rs1326821916 (GAK) are shown in Supplementary Fig. 8. All three genes have low tissue specificities, with almost equal gene expression across different tissue types (Supplementary Fig. 9). ASH1L and GAK were previously reported to be associated with obesity traits such as BMI and waist-hip ratio in GWAS of UK Biobank[39]. Obesity traits were shown to have significant genetic correlation and are causally associated with glycaemic traits in our pairwise genetic correlation analysis and MR. Furthermore, Klarins et al. reported that GAK was associated with lipid traits (HDL and TG) in their genome-wide meta-analysis. Lipid traits were also shown to have significant genetic correlation and causally associated with glycaemic traits in our pairwise genetic correlation analysis and MR[40]. All these implied that ASH1L and GAK are crucial cardiometabolic genes in T2D. Inhibition of RAD21 has been demonstrated to increase insulin secretion in a MIN6 mouse beta cell line[17] and RAD21 was associated with reduced hematopoietic stem cell self-renewal in aging and inflammation[41]. Rare mutation of RAD21 has been reported in Cornelia de Lange syndrome with premature physiological aging and gastrointestinal tract difficulties[42]. However, RAD21 has never been reported to be linked to any cardiometabolic traits in any GWAS. Further functional work in both cellular and animal model will be required to confirm the role of RAD21 in T2D and the link between insulin secretion, physiological aging and T2D.

As discussed in ref. [43], it is more ideal to consider the ancestry-trait-specific Bonferroni-corrected significance threshold. In our study, we only consider the Taiwanese population, and the maximum number of tested SNPs is 5,981,581 for all traits. Therefore, the most stringent ancestry-trait-specific Bonferroni-corrected significance threshold would be $8.36 \times 10^{-9}$. Among the three highlighted genes that were identified based on the traditional threshold $5 \times 10^{-8}$ only the gene HACL1 (rs1481559294)-HbA$_{1c}$ trait with $P$ value $= 4.24 \times 10^{-8}$ larger than the most

stringent Bonferroni-corrected significance threshold in our study. However, as mentioned above that the study of Hacl1 knockout mice supports the potential involvement of *HACL1* for fatty acid, we believe that *HACL1* is still a promising candidate gene for metabolic traits.

Our absolute risk model for T2D using PRS and risk factors should be applicable for any Asian population. The strengths of absolute risk model for T2D are the combination polygenic risk of SNPs and risk factor measurements for all subjects allowed joint evaluation on the effects of PRS and family history of T2D. Our findings could be useful in global efforts to generate trans-ancestry PRS. The good model calibration of relative risk demonstrated the validity of the model. The main limitation of our PRS model is that self-report disease status often under-estimates the true disease prevalence/incidence. As TWB is an on-going project, we expect to have a sufficiently large follow-up dataset on T2D in the near future, which will allow us to validate our prediction model and evaluate its applicability for population-wide screening on T2D to identify high-risk individuals for early intervention.

## Methods
**Study population**. We used individual genotype and phenotype data of subjects recruited from 2012 to 2019 of Taiwan Biobank (TWB) for subsequent data analysis. (https://www.twbiobank.org.tw/) The population in Taiwan consists of mostly East Asian ancestry, specifically Han Chinese, therefore, they are suitable to serve as a representative study sample for Asian population. Detailed information of TWB dataset is available in the previous publication[23]. This study has been approved by the internal review board of the Academia Sinica (Num: AS-IRB02-109063) and the research ethics committee of Taiwan University Hospital (No. 201507020RINB), and Taiwan Biobank. All participants gave informed consent when joining TWB, which allows for sharing of all anonymized data with authorized researchers. Participants can withdraw consent to sharing their data at any stage of their participation in TWB.

**GWAS**. We conducted GWAS through logistic regression model (for binary traits) and linear regression model (for continuous traits) under the assumption of additive allelic effects of the SNP dosages via PLINK v2.0. The regression models were adjusted for age, gender and the first ten genetic principal components.

**Genotyping and imputation**. Detailed genotyping and imputation procedures have been described earlier[23]. For this study 95,252 subjects were genotyped with either the customized TWB1 array ($N_{TWB1}$ = 27,737 DNA variants) or TWB2 array ($N_{TWB2}$ = 68,978) or both ($N_{both}$ = 1496) and the last group was also typed by whole genome sequencing (WGS).

**Quality control**
*Binary traits*. We first homogenized control individuals by removing comorbid individuals for each trait. Comorbid diseases are defined by a data-driven method using the Partitioning Around Medoids (PAM)[44] algorithm in the cluster package of R (version 3.6) and φ-correlation as our distance matrices. Best-fit group numbers were selected by maximizing the silhouette score[45]. The final sample sizes for each trait are shown in Supplementary Data 1.

*Quantitative traits*. For each trait, outliers beyond three standard deviations (two-tailed) were excluded. Individuals with missing values in any trait were dropped from the analysis. A total of 34 quantitative traits were used in this study (Supplementary Data 1).

*Genotype data*. The genotype data from the TWB1 and TWB2 arrays were merged using the GRCh38 assembly and annotation provided by TWB. After filtering out samples with a call rate <0.99 or sex mismatch in either of the TWB1 and TWB2 datasets, 95,215 samples and 95,673 variants remained. For kinship estimation, computation of principal components (PCs), and genomic relation matrix, SNPs were extracted by the following criteria: (1) SNP IDs, chromosomes, physical positions, minor alleles and major alleles had to be all identical in both datasets; (2) call rate >0.95; (3) MAF > 0.01; (4) deviation from Hardy–Weinberg equilibrium (HWE) $P$ > 0.001; and (5) no INDELs. For sample filtering, arrays with generated genotypes for <95% of the loci were excluded. PLINK v1.9 software was used to identify samples with genetic relatedness indicating that they were from the same individual or from first-, second- or third-degree relatives. These determinations were based on evidence for cryptic relatedness from identity-by-descent status (pi-hat cutoff of 0.125). After removing first-, second- and third- degree relatives, 77,072 independent samples and 59,521 SNPs remained.

*Imputation data*. For our analysis, we merged the imputed TWB1 and TWB2 genotype data and selected SNPs according to the merged imputation data released by TWB. Low-quality variants were filtered out using PLINK if an SNP met any of the following criteria: (1) MAF ≤ 0.001; (2) imputation INFO score ≤ 0.8; (3) call rate ≤ 0.95; and (4) deviation from HWE ($P ≤ 10^{-10}$). Supplementary Table 1 shows the number of SNPs included in the analyses for each trait.

*For building PRS model*. Additional typical QC procedure for SNPs to build PRS models were applied. Multiallelic SNPs and SNPs with ambiguous strands were removed from the analysis. SNPs with MAF ≤ 0.01, low imputation quality (info <0.3) or deviation from HWE ($P < 10^{-6}$) were also excluded.

**Statistics and reproducibility**
*PheWAS*. Significant signals for all binary traits were first screened using a genome-wide significance threshold of $P ≤ 5.0 × 10^{-8}$ with PLINK2 (https://www.cog-genomics.org/plink/2.0). Linear regression models were used to evaluate the association of all SNPs with each of the 34 quantitative traits under the assumption of additive allelic effects by PLINK. Unless described otherwise, both binary and quantitative traits were adjusted for age and sex with the first ten principal components (PCs) estimated by EIGENSOFT (version 6). Since the standard threshold of $5 × 10^{-8}$ had been used in many PheWASs such as in homogeneous population[20] and also in trans-ancestral analysis[29], therefore, we set the standard threshold of $5 × 10^{-8}$ for our GWAS.

*Conditional association analysis*. Conditional analyses were performed for each aforementioned defined locus using the PLINK2 "--condition" flag. Association tests were conducted based on the generalized linear model with adjustment for all covariates listed in the PheWAS section and an additional lead SNP genotype. Linkage disequilibrium (LD) was estimated with the pairwise squared correlation ($r^2$) within each locus with a window size of 4000 SNPs using 1496 TWB WGS data. SNP was considered to be independent of the lead SNP if $r^2 ≤ 0.1$.

*Gene mapping and functional annotation*. Post-GWAS analysis of gene mapping, functional annotation, and tissue-expression analysis of prioritized genes was conducted using FUMA SNP2GENE and GENE2FUNC functions[46]. Independent significant SNPS are defined as $P < 5 × 10^{-8}$ and $r^2 < 0.6$, lead SNPs if pairwise SNPs had $r^2 < 0.1$. The maximum distance between LD blocks to merge into a genomic locus was set to 250 kb. The genetic data of East Asian populations in 1000 G phase 3[47] were viewed as reference data to conduct LD analyses. Gene expression of different tissues was estimated with gene expression data of 54 tissue types from GTEx v8[48]. Consensus transcript expression levels for *HACL1* and *GAD2* in 55 human tissues were also generated from Protein Atlas based on transcriptomics data from the two sources HPA and GTEx[49].

*Phenotype–phenotype genetic correlation*. We used bivariate LD score regression[1] to calculate genetic correlations between all phenotype pairs. TWB WGS data of 1496 samples were used to compute LD scores with a one cM window size. Summary statistics of the SNPs passing QC criteria described in the above methods section were utilized to perform this calculation. The false discovery rate (FDR) calculated by the Benjamini–Hochberg method[50] was used to adjust for multiple testing of 946 combinations of pairs of traits using the Python module *statsmodels* (www.statsmodels.org). A significant genetic correlation between phenotypes was considered if FDR ≤ 0.05. Data visualization was performed through the ldsc-corrplot-rg script (https://github.com/mkanai/ldsc-corrplot-rg) and R.

*Mendelian randomization*. Mendelian randomization (MR) methods utilize common genetic variants to estimate the causal relationship of risk factors with disease outcomes[2,51]. In our MR analysis, we first filtered horizontal pleiotropic instrumental variables (IVs)[52] using MR-Egger[53] and identified outliers through MR-PRESSO[54]. Finally, the valid IVs were applied through IVW[55] to analyze causal relationships between the following eleven clinical measurements as exposures (i.e., BMI, waist circumference, body fat percentage, waist-hip ratio, hip circumference, triglyceride, HDL-C, VLDL-C), and outcomes (i.e., three glycemia-related traits: T2D, FG, and HbA$_{1c}$).

TWB samples was randomly split into two sub-groups: 3/4 (G group) for GWAS and the remaining 1/4 (MR group) was tested using the IVW method for MR analysis. For the IV selection, GWAS were carried out in the same manner described above using the G group, and significant SNPs for the eight exposures were selected. The causal associations of the eight exposures with T2D were investigated using the MR group with logistic regression models. The resulting associated SNPs (IVs), their regression coefficients and the effect estimates of the exposures on the outcome were obtained by pooling all MR estimates using the random-effects IVW method. To ensure independence of IVs, strict clumping was performed with a $r^2$ threshold of 0.01 and physical distance threshold of 10 Mb through clump command of the TwoSampleMR package[56] in R.

*Data preprocessing for absolute risk modeling*. To build absolute model for T2D risk estimation based on PRS and risk factors in T2D-free individuals, we divided the process into three stages: construction of PRS, absolute risk modeling, and

validation analysis. For the data preprocessing, we first identified individuals who were free of T2D at baseline and with more than one visit records, which was 15,664 individuals, to be utilized subsequently in the validation analysis (TWB-for-val). For the PRS model construction, we randomly selected one third of the remaining TWB samples, excluding the 15,664 individuals for validation analysis (TWB-for-PRS). The remaining 2/3 samples (TWB-for-AR) was used to build the absolute risk model.

*Constructing polygenic risk score (PRS) model.* A PRS was calculated as a weighted sum of the number of alleles of SNPs. To estimate the weights for PRS models, we used GWAS summary statistics from BBJ: estimates of regression coefficients ($\hat{\beta}_j$), their standard errors ($\hat{\sigma}_j$), and associated $P$ values ($p_j$) for each SNP $j$. The QC procedure for SNPs is available in methods.

To calculate PRS, two methodologies were utilized. The first method was the standard clumping and thresholding (C + T) method. The hyperparameters for this method were the thresholds for the correlation $r^2$ and $P$ value $p$. The parameter spaces were the Cartesian product of $r^2$ and $p$, where $r^2 \in \{0.01, 0.1\}$ and $p \in \{0.05, 0.001, 0.005, 1e^{-4}, 5e^{-4}, 1e^{-5}, 5e^{-5}, 5e^{-6}\}$. For each pair of ($r^2$, $p$), we used PLINK with a window size of 10 Mb to select SNPs. For model selection, we used the TWB-for-PRS sample to choose optimal tuning parameters. The second method was the C + T method with winner's curse correction proposed by Shi et al.[57] and the same strategy to select an optimal PRS model.

*Absolute risk modeling and validation analysis for T2D.* To build the absolute risk model, we utilized the R package iCARE[58] to project the individual risk. We used the best PRS model described above and transformed it into 4 quintile variables to facilitate interpretation, where the first quintile represents the lowest 20% of the PRS sample in our sample population and so forth for the other quintiles. Similarly, the continuous BMI values were categorized into four classes (BMI < 18.5, 18.5 ≤ BMI < 24, 24 ≤ BMI < 28, 28 ≤ BMI). To apply iCARE for absolute risk estimation, relative risk (RR) estimates is required for all variables in the model, including PRS quintiles, categorical BMI, and family history of T2D. We used TWB-for-AR samples to estimate odds ratio (OR) associated with the aforementioned variables by logistic regression. As the prevalence of T2D in the Taiwanese population is about 10%, the OR gives a reasonable approximation to the RR. Absolute risk estimation was also based on the age and sex-specific T2D incidence rate and mortality rate from causes other than T2D in the Taiwanese population as recommended in ref. [9] and data from the Taiwan Ministry of Health and Welfare (https://www.mohw.gov.tw/mp-2.html). Note that the data on the incidence and mortality rates from these references are based on diabetes mellitus cases; however, they should provide a reasonably good approximation to T2D as T2D represents the majority (>95%) of diabetes mellitus cases in our data.

Lastly, we used the R package iCARE to conduct validation analysis on TWB-for-val sample. Model calibration was assessed by comparing the model projected absolute and relative risk estimates to the observed values in the TWB-for-val sample. The Hosmer-Lemeshow and Chi-square tests were used to judge goodness of model fit, respectively. In addition, the model discrimination was assessed by area under the curve (AUC).

**Reporting summary**. Further information on research design is available in the Nature Portfolio Reporting Summary linked to this article.

## Data availability

GWAS summary statistics of T2D, HbA1c, and Fasting Glucose have been provided to the NHGRI-EBI GWAS Catalog, and the study accession numbers are GCST90161239, GCST90161237, GCST90161236. Summary statistics were downloaded from the UKB Biobank (UKBB) and the Biobank Japan Project. The Biobank Japan Project: Summary statistics of T2D, HbA1c, and Blood sugar were acquired from the Biobank Japan Project website (http://jenger.riken.jp/en/result). The UK Biobank: Summary statistics of T2D, HbA1c and Fasting Glucose were acquired from Neale's lab website (GWAS round 2) (http://www.nealelab.is/uk-biobank). Supplementary Data 4 contains source data underlying Fig. 1a. Supplementary Data 15 contains source data underlying Fig. 4. Other data used in this study were obtained from Taiwan Biobank, which is publicly available on request, while we are not authorized to redistribute the data. Analysis results can be shared on request by contacting the corresponding authors for reasonable use.

## Code availability

No custom computer code was used in this study. We used publicly available software (URLs listed below) in this research. Genetic association analyses were performed using PLINK2 (https://www.cog-genomics.org/plink/2.0). The Mendelian Randomization analyses were done using the R package MendelianRandomization (https://cran.r-project.org/web/packages/MendelianRandomization/index.html). Polygenic risk scores were calculated using the software plink (https://www.cog-genomics.org/plink/) and R programming (https://www.r-project.org), and absolute risk estimation was conducted by R package iCARE (https://www.bioconductor.org/packages/release/bioc/html/iCARE.html). SNP heritability and

genetic correlations were estimated using LD score regression (https://github.com/bulik/ldsc) and LD hub (http://ldsc.broadinstitute.org/). Functional annotations were done using FUMA (https://fuma.ctglab.nl/). LocusZoom (https://github.com/Geeketics/LocusZoms). UpSet plot (https://github.com/hms-dbmi/UpSetR). Liftover (https://genome.sph.umich.edu/wiki/LiftOver).

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

## Acknowledgements

We acknowledge the staff at Taiwan biobank for their hard work in collecting and distributing the data, and give special thanks to Dr. Te-Chang Lee. We also acknowledge Benjamin Neale's team, UK Biobank, and the Biobank Japan Project for making their analysis results publicly available. This work was supported by the following grants: MOST 109-2314-B-001 -006 -MY2 and MOST 111-2314-B-001.008. We thank the Institute of Biomedical Sciences, Academia Sinica of Taiwan, and the National Science and Technology Council of Taiwan.

## Author contributions

Conceptualization: C.-J.L., T.-H.C., A.-R.H., and C.S.-J.F.; Data curation: C.-J.L., J.-J.S., C.-C.C., S.-J.W., and C.-L.H.; formal analysis: C.-J.L., T.-H.C., A.M.-W.L., J.-J.S., C.-C.C., S.-J.W., C.-L.H.; funding acquisition: C.S.-J.F.; software: C.-J.L.; supervision: A.-R.H., W.-S.Y., and C.S.-J.F.; visualization: C.-J.L., T.-H.C., A.M.-W.L., and J.-J.S.; writing—original draft: C.-J.L., T.-H.C., A.M.-W.L., A.-R.H., W.-S.Y., C.S.-J. F.; writing—review & editing: S.-W.C.; biological interpretation: A.M.-W.L., P.-L.C., and W.-S.Y.

## Competing interests

The authors declare no competing interests.
