## [Peer Review File · Communications Biology]

Reviewers' comments:

Reviewer #1 (Remarks to the Author):

Lee, et al, present the first genome-wide PheWAS of a group of binary and quantitative traits relevant to type 2 diabetes and accompanying glycemetic traits in the Taiwanese BioBank study population. They perform several complementary analyses to identify previously unreported GWAS genes for multiple traits, and to determine whether anthropometric or lipid traits may be causally associated with glycemetic traits using Mendelian randomization. Finally, they use a split-sample design to calculate a PRS for T2D stratified by sex and BMI. This study provides a new take on a well known group of phenotypes and traits that affect a large proportion of people. Of particular interest is the potential clinical applicability of these findings to the larger Taiwanese or Han Chinese population, given studies showing that PRS developed using European-ancestry GWAS often lack portability to other populations, diminishing their utility.

Major comments:

(1) The authors highlight several genes of interest but could characterize them better within this study population. Do these genes have eQTLs in tissues relevant to glycemetic traits even in broadly expressed genes and is there colocalization with the authors' lead GWAS SNPs? Do the allele frequencies of lead variants in this study differ from GWAS in similar or different study populations? Do knockout/knockdown mouse models of these genes exist and if so, do they have relevant phenotypes? Regional association plots (Locus-Zoom or similar) or UCSC genome browser plots with relevant tracks should be shown for all genes of interest described in the discussion. If any are particularly interesting, one could be highlighted as a main figure with the remaining as supplemental figures.

(2) Related to the previous comment, in the conclusion the authors mention the importance of functional annotation, but they do not talk about it at all in the results section other than briefly mentioning gene expression of genes identified using FUMA. It would be useful, particularly in a population that is traditionally underrepresented in GWAS, to characterize the lead SNPs in terms of function with regard to their mapped genes (i.e., in promoter region, coding variants, etc) and actually functionally annotate them, for example to identify whether transcription factor binding sites are disrupted or some other effect is predicted (even CADD scores) for all GWAS significant genes but especially previously unreported genes. The authors should also clarify whether they performed any fine-mapping analysis to determine the best candidate causal variants or only used the lead GWAS SNP for all downstream analyses.

(3) A Bonferroni correction for the number of traits and SNPs tested would be too conservative, but the field standard for testing millions of SNPs across multiple traits is typically 5×10^{-9} . If the authors use the standard adjustment for a single-trait GWAS, please justify in the methods.

(4) In the MR analyses, adjustment for multiple testing of 24 different analyses would give a significance threshold of $0.05/24 = 0.0021$, which would eliminate several of the findings labeled as significant in the Results section. If the authors believe a threshold of 0.05 is appropriate despite performing multiple tests, please justify and cite an example for this being standard in the field.

(5) Figure S2 should be the main figure, and traits that were analyzed for MR should be annotated in some way (boxes have a colored border or trait labels have an asterisk or something). An interpretation of the seeming disparity between two traits having shared heritability in the bivariate LDSC but not contributing to horizontal pleiotropy in the MR analyses would be useful.

(6) One great benefit of this study population is the presence of variants that are monomorphic or very rare in other ancestral populations. The paper and the field would benefit from a brief comparison

of allele frequencies of lead variants in this study vs other published GWAS like UKBB or BBJ, and whether any of the discovery genetic associations were driven by ancestry-specific variants.

Minor comments:

(1) This sentence is confusing: "In our study, lipid profiles such as HDL-C and VLDL-C were found to be superior to anthropometric measurements in predicting the risk of glycemia-related traits." BMI performed better than VLDL. Furthermore, HDL and VLDL are not anthropometric measures but lipid traits. Please clarify this sentence and the reason for choosing VLDL over BMI.

(2) The supplemental tables PDF is not legible and only appears to contain parts of two tables. If supplementary material uploaded as a PDF for resubmission, please double-check that all files are easy to read. The Excel file had no issues.

(3) Figure 3 A/B panels the differently sized but same shape representations for the PRS quintiles is confusing. Either having different colors for the four groups (BMI x sex) or having one color for the top PRS quintile and a different color for the bottom quintile with the four different shapes would make comparisons much easier for the reader. Figures 3C-F can be in the supplement.

(4) In lines 96-97 of the Methods and line 398 on the Discussion the authors should clarify whether their results are applicable to all Asian populations (including South Asian for example) or best suited to Taiwanese or Han Chinese-ancestry populations.

(5) Given the emphasis on PRS and its potential interpretation/application, the introduction should include one or two sentences on the epidemiology of T2D in Taiwan--prevalence in the adult population, average age of onset, and maybe estimated effect on general health or productivity lost to this disease (e.g., DALYs or annual national cost of treatment)

(6) In the first sentence of the introduction "Mendelian" is misspelled

Reviewer #2 (Remarks to the Author):

In the current study, Lee et al. performed a phenome-wide association study (pheWAS) for 10 diseases and 34 quantitative traits in the Taiwan Biobank (TWB). They also conducted further conditional analyses, genetic correlation and a Mendelian randomization study. Overall, they identified four novel genetic variants within or close to the HACL1, RAD21, ASH1L and GAK genes. Many of the identified loci were overlapping reflecting high pleiotropy and/or high correlation between the traits tested. It is a straightforward study with sound methodology, nevertheless, some major points both should be clarified.

Major comments

1. How did you perform GWAS for binary traits? Neither method nor supplemental methods include details about tests and models. What kind of test were performed for association? These are crucial steps; therefore, they should be included in detail.

2. Why did you select the first ten principal components, age and sex as covariates? Are these covariates associated with the traits? StepAIC or a similar approach can be used to select the models for each trait (<https://ashutoshtripathi.com/2019/06/10/what-is-stepaic-in-r/>).

3. Genome-wide significance threshold ($P \leq 5.0 \times 10^{-8}$) was defined in the European population. Would it be different in Taiwanese cohort? Please see the relevant study showing different significance level

for different genetic ancestries:

Smith et al, 2022. [https://www.cell.com/ajhg/fulltext/S0002-9297\(22\)00101-X](https://www.cell.com/ajhg/fulltext/S0002-9297(22)00101-X)

Based on the new genome-wide significance level, the number of genome-wide significant loci would change for each trait.

4. Line 170; "Multi allelic SNPs and SNPs with ambiguous strands were removed from the analysis." Were these filters applied for other pheWAS analyses or specific for only PRS? In addition, filters for MAF, HWE and imputation quality were presented in PRS section. Please make sure that all filters and analyses included in methods/or supplemental method (if it is same for all analyses, you could include one general method, or separate for each analysis if the filters for specific for each analysis).

5. Supplemental method: What is the rationale of setting a minimum HWE p-value of 0.001 for genotyping data and HWE ($P \leq 10^{-10}$) for imputed data?

Minor comments:

1. In lines 42 and 61, please correct the spelling for Mendelian randomization. In Line 66, UKBB, one B letter is missing.

2. In abstract "In total, we identified 995 significantly associated loci with more than 100 novel loci, specific to Taiwanese population. Furthermore, our analysis highlighted the genetic pleiotropy of loci associated with complex disease and associated quantitative traits. To demonstrate the versatility of our PheWAS, further extensive analysis on glycemic phenotypes (T2D, fasting glucose and HbA1c) was performed and identified 115 significant loci"

2a. Please specify the exact number of more than 100 novel loci

2b. Did you identify 995 "variants" or "loci" in more than 100 loci?

2c. How did you figure out that these variants specific to Taiwanese population? Do not they exist in other populations? How did you determine this specificity?

3. In lines 95 and 338, genetic ancestry could be a better term or description instead of ethnicity. Ethnicity is a socially constructed concept and is used to refer to groups of people who shares a similar history and culture, such as language or history. Therefore, although it may be correlated with genetic variation, it should not be used as a proxy for genetic differences among groups.

4. In supplemental materials, lines 22-27: how did you perform the kinship analysis and filtering? It can be clarified by including the tools and filtering thresholds for relatedness.

5. Line 47, "significant loci with four novel genetic variants (HACL1, RAD21, ASH1L and GAK)": Are the variants within these genes, or are these the closest genes to the significant hits?

Reply to Reviewers

Reviewer #1 (Remarks to the Author):

Lee, et al, present the first genome-wide PheWAS of a group of binary and quantitative traits relevant to type 2 diabetes and accompanying glycemic traits in the Taiwanese BioBank study population. They perform several complementary analyses to identify previously unreported GWAS genes for multiple traits, and to determine whether anthropometric or lipid traits may be causally associated with glycemic traits using Mendelian randomization. Finally, they use a split-sample design to calculate a PRS for T2D stratified by sex and BMI. This study provides a new take on a well known group of phenotypes and traits that affect a large proportion of people. Of particular interest is the potential clinical applicability of these findings to the larger Taiwanese or Han Chinese population, given studies showing that PRS developed using European-ancestry GWAS often lack portability to other populations, diminishing their utility.

Major comments:

(1) The authors highlight several genes of interest but could characterize them better within this study population. Do these genes have eQTLs in tissues relevant to glycemic traits even in broadly expressed genes and is there colocalization with the authors' lead GWAS SNPs? Do the allele frequencies of lead variants in this study differ from GWAS in similar or different study populations? Do knockout/knockdown mouse models of these genes exist and if so, do they have relevant phenotypes? Regional association plots (Locus-Zoom or similar) or UCSC genome browser plots with relevant tracks should be shown for all genes of interest described in the discussion. If any are particularly interesting, one could be highlighted as a main figure with the remaining as supplemental figures.

Response: We appreciate the reviewer's reminder. We have added additional information on eQTL, tissue expression and colocalization of glycemic trait related genes including the highlighted genes of interest in Supplementary Tables S11-13 and Figures S3 & S4.

Supplementary Tables S11-13 are available at the following link due to the size of the file:
http://www.csjfann.ibms.sinica.edu.tw/eag/PHEWAS/PHEWAS_supp.html

Summary of SNPs and mapped genes			
	T2D	HbA1c	Fasting Glucose
#Genomic risk loci	7	26	29
#lead SNPs	7	40	41
#Ind. Sig. SNPs	9	88	106
#candidate SNPs	302	2771	4142
#candidate GWAS tagged SNPs	168	1219	2075

#mapped genes	65	440	615
---------------	----	-----	-----

Figure S3. Summary of FUMA SNP2GENE for GWAS of T2D (top), HbA1C (middle) and fasting glucose (bottom).

Figure S4. Differentially expressed genes of GTEx v8 54 tissue types with genes mapped from GWAS summary statistics of T2D (top), HbA1C (middle) and fasting glucose (bottom).

Only two out of the six SNPs have significant eQTL in tissues relevant to glycemic traits which is rs6547629 (*GCKR*) in thyroid tissues and rs61839365 (*GAD2*) in pancreatic tissues.

The lead SNP rs6547692 for *GCKR* gene has similar prevalence across TWB (MAF = 0.485, BBJ (MAF = 0.4387) and UKBB (MAF = 0.44). From Genome Aggregation Database

(gnomAD) [1], we noticed that rs1481559294 for *HACLI* was a relatively rare variant (African n=42024 MAF = 0.00002, East Asian n=3132 MAF = 0.0006) as compared to the MAF in our population of 0.0013 (n=77,702). The lead SNP rs61839365 for *GAD2* is more prevalent in East Asian with MAF of 0.293 in TWB and 0.416 in BBJ as compared to 0.18 in UKBB.

The ancestry-specific allele frequencies and phenotypic relevance as above are explained in discussion section on page 18, paragraph 1, page 18, paragraph 2, and page 19, paragraph 2.

To the best of our knowledge, only *HACLI*, *ASHIL* and *GAK* genes have relevant animal models. *HACLI* knockout mouse model alone has been reported to be associated with metabolic traits (fatty acid). The following statement has been added to the discussion section on page 18 paragraph 2.

“Jenkins *et al* [2] showed that *Hac1l* KO mice had lower plasma and liver C17:0 fatty acid, but did not observe significant difference in adipose tissue.”

Gene	SNPID	Chr:position (GRCh38)	CADD \$	eQTL p-value	tissue
GCKR ENSG00000084734	rs6547692	2:27512105	1.148	1.6e-11	Thyroid
HACLI ENSG00000131373	rs1481559294	3:15584876	0.602	No Data Available	
GAD2 ENSG00000136750	rs61839365	10:26226737	1.782	0.000042	Pancreas
RAD21 ENSG00000164754	rs11994747	8:116848407	1.168	No significant eQTLs were found	
ASHIL ENSG00000116539	rs371382391	1:155420617	1.202	No significant eQTLs were found	
GAK ENSG00000178950	rs1326821916	4:860479	3.280	No Data Available	

Furthermore, regional association plots of *GCKR*, *HACLI* and *GAD2* have been added as the main figure (Figure 5) and regional plots (before and after conditional analysis) of *RAD21*, *ASHIL* and *GAK* have been added to supplementary figures (Figure S8).

Figure 5. Regional association plots, identified by comparison of Taiwan Biobank (TWB), Japan Biobank (BBJ) and UK Biobank (UKBB) genome-wide association studies (GWASs). The X-axis represents the position of loci (hg19). The Y-axis represents $-\log_{10}(P)$. Red dots are p values of variants from TWB. Green dots are p values from the BBJ. Blue dots are p values from UKBB. The red horizontal line represents $P = 5 \times 10^{-8}$. The black dashed line represents the location of the gene. A) *GCKR* gene; B) *HACLI* gene and C) *GAD2* gene.

Supplementary Figure S8. Regional plots before (left) and after (right) a conditional analysis. A) *RAD21* gene; B) *ASH1L* gene and C) *GAK* gene.

A

(2) Related to the previous comment, in the conclusion the authors mention the importance of functional annotation, but they do not talk about it at all in the results section other than briefly mentioning gene expression of genes identified using FUMA. It would be useful, particularly in a population that is traditionally underrepresented in GWAS, to characterize the lead SNPs in terms of function with regard to their mapped genes (i.e., in promoter region, coding variants, etc) and actually functionally annotate them, for example to identify whether transcription factor binding sites are disrupted or some other effect is predicted (even CADD scores) for all GWAS significant genes but especially previously unreported genes. The authors should also clarify whether they performed any fine-mapping analysis to determine the best candidate causal variants or only used the lead GWAS SNP for all downstream analyses.

Response: We appreciate the reviewer’s comment and acknowledge the insufficient explanation on the functional annotation of relevant genes. Additional paragraphs have been added to both results and discussion to reflect this. As our main research objective is to determine genetic variants unique to Taiwan Han Chinese, we first filtered out TWB specific variants by comparing with other biobanks (UKBB & BBJ). We noticed that there were only a small number of TWB specific variants (none for T2D GWAS; two each for FG and HbA1C GWAS), no fine mapping was done to identify best candidate causal variant. However, in our subsequent downstream analysis, we did some of components in fine-mapping for our TWB unique variants, such as colocalization and functional annotation. We will collaborate and perform future studies to further elaborate on these specific loci.

The following statement has been added to the Result section on page 15, paragraph 1.

“Summary of SNPs and mapped genes by FUMA SNP2GENE are shown in Supplementary Tables S11-13 and Figure S4. From the SNPs identified, most of them are located in intronic and intergenic region; only minuscule number of SNPs are located in the exonic region (T2D: 0.3%, HbA1c: 1.4%, Fasting glucose: 0.8%) (Supplementary Figure S3). The mapped genes were not statistically significantly expressed in any specific tissue types (Supplementary Figure S4). However, we noticed up-regulation of differentially expressed genes in glycaemic-related tissue types such as kidney, pancreas, stomach and adipose tissues. Summary of the functional annotation of the mapped genes are available in Supplementary Tables S11-13 and Figures S3 & S4. (Please refer to response to comment 1 for the supplementary table S11-S13 and Figure S3 & S4)

(3) A Bonferroni correction for the number of traits and SNPs tested would be too conservative, but the field standard for testing millions of SNPs across multiple traits is typically 5×10^{-9} . If the authors use the standard adjustment for a single-trait GWAS, please justify in the methods.

Response: We agree that Bonferroni correction in this case might be too stringent and conservative with over exaggeration in false negative results. Since standard threshold of 5×10^{-8} had been used in many PheWASs such as in homogeneous population [3] and also in trans-ancestral analysis [4], therefore, we set the standard threshold as 5×10^{-8} .

The above sentences have been added in the methods section page 7 paragraph 1.

(4) In the MR analyses, adjustment for multiple testing of 24 different analyses would give a significance threshold of $0.05/24 = 0.0021$, which would eliminate several of the findings labeled as significant in the Results section. If the authors believe a threshold of 0.05 is appropriate despite performing multiple tests, please justify and cite an example for this being standard in the field.

Response: Thanks for reviewer’s reminder. We agree that Bonferroni-corrected significance threshold of $p=0.0021$ ($0.05/24$) is preferred in order to adjust for multiple testing however, it might be too conservative according to Tian *et al* (4). Therefore, we regarded any associations with p-values between 0.05 and 0.0021 as suggestive evidence of possible association [5].

We have now added the paragraph below into the Result section on page 14, paragraph 1.

“In the two-sample MR analysis on T2D, the overall causal estimate (IVW odds ratio (OR) estimate) for T2D per unit increase in BMI was 1.2566 ($P = 0.0011$), for the effect of a 1-unit increase in HDL-C on the risk of T2D was 0.9762 ($P = 0.0087$), and for the effect of a 1-unit increase in VLDL-C on the risk of T2D was 0.9832 ($P = 0.01$) (Table S8). In the two-sample MR analysis on HbA1c, the overall causal estimate for HbA1c per unit increase in BMI was 1.0317 ($P = 0.0071$) (Table S9). As for the two-sample MR analysis on FG, the overall causal estimate for FG per unit increase in HDL-C was 0.9472 ($P = 0.0229$) (Table S10). By contrast, WC, FR, WHR, HC, and triglyceride were not found to be significantly associated with T2D, HbA1c, and FG (Table S8-Table S10). After the Bonferroni correction ($p = 0.0021$ ($0.05/24$)), the causal relationship between BMI and T2D remained from our MR analyses.”

(5) Figure S2 should be the main figure, and traits that were analyzed for MR should be annotated in some way (boxes have a colored border or trait labels have an asterisk or

something). An interpretation of the seeming disparity between two traits having shared heritability in the bivariate LDSC but not contributing to horizontal pleiotropy in the MR analyses would be useful.

Response: Thanks for reviewer’s reminder and suggestion. We concur that Figure S2 should be the main figure to reflect the importance of this finding. Following that, we have added a new Figure 3 to substitute the original Figure S2 which we believe will be better in helping readers visualize the finding.

Figure 3. Overlapping loci within glyceamic traits. The upset plot [6] summarizes shared loci of three glyceamic traits. The dot-and-line chart in the bottom combination matrix indicates intersections between three traits. For example, the second column from the left indicates set of loci only associated with HbA1c, while the middle column indicates loci associated with both HbA1c and FG. The upper bar chart shows the number of associated loci in each set. The lower left horizontal bar chart represents number of loci associated with T2D (7), HbA1c (26) and FG (29).

We are sorry for the confusion that “seeming disparity between two traits having shared heritability in the bivariate LDSC but not contributing to horizontal pleiotropy in the MR”. Actually, no horizontal pleiotropy assumption is important for MR models (Davey Smith, G. & Hemani, G, 2014). We noticed that in our original analysis, rs654769 contributed to pleiotropy between FG and VLDL-C, therefore it was deleted for the new analyses. The results found no causality between VLDL-C and FG. The revised MR analysis is shown in Table S10.

(6) One great benefit of this study population is the presence of variants that are monomorphic or very rare in other ancestral populations. The paper and the field would benefit from a brief comparison of allele frequencies of lead variants in this study vs other published GWAS like UKBB or BBJ, and whether any of the discovery genetic associations were driven by ancestry-specific variants.

Response: Thanks for reviewer's reminder. Increasing the population diversity for genetic studies especially for under-represented minorities is one of the key objectives of Taiwan Biobank. The lead SNP rs6547692 for *GCKR* gene has similar prevalence across TWB (MAF = 0.485, BBJ (MAF = 0.4387) and UKBB (MAF = 0.44). From Genome Aggregation Database (gnomAD) [1], we noticed that rs1481559294 for *HACL1* was a relatively rare variant (African n=42024 MAF = 0.00002, East Asian n=3132 MAF = 0.0006) as compared to the MAF in our population of 0.0013 (n=77,702). The lead SNP rs61839365 for *GAD2* is more prevalent in East Asian with MAF of 0.293 in TWB and 0.416 in BBJ as compared to 0.18 in UKBB.

The ancestry-specific allele frequencies and phenotypic relevance as above are explained in discussion section on page 18, paragraph 1, page 18, paragraph 2, and page 19, paragraph 2.

Minor comments:

(1) This sentence is confusing: "In our study, lipid profiles such as HDL-C and VLDL-C were found to be superior to anthropometric measurements in predicting the risk of glycemia-related traits." BMI performed better than VLDL. Furthermore, HDL and VLDL are not anthropometric measures but lipid traits. Please clarify this sentence and the reason for choosing VLDL over BMI.

Response: We apologize for the confusion and have removed the sentence in our revised manuscript.

(2) The supplemental tables PDF is not legible and only appears to contain parts of two tables. If supplementary material uploaded as a PDF for resubmission, please double-check that all files are easy to read. The Excel file had no issues.

Response: Thanks for reviewer's reminder. We have uploaded supplementary materials with higher quality for better readability.

(3) Figure 3 A/B panels the differently sized but same shape representations for the PRS quintiles is confusing. Either having different colors for the four groups (BMI x sex) or having one color for the top PRS quintile and a different color for the bottom quintile with the four different shapes would make comparisons much easier for the reader. Figures 3C-F can be in the supplement.

Response: Thanks for the reviewers' suggestions. We have modified Figure 4 and moved Figures 3C-F to the supplementary materials accordingly.

(4) In lines 96-97 of the Methods and line 398 on the Discussion the authors should clarify whether their results are applicable to all Asian populations (including South Asian for example) or best suited to Taiwanese or Han Chinese-ancestry populations.

Response: Thanks for reviewer’s reminder. To a certain degree, we believe that our PRS model might be applicable to East Asian populations in contrast with PRS model developed from European ancestry. Wei *et al* [7] showed that TWB cohort represents diverse ancestry of different province of mainland China and 99% of TWB cohort are Han Chinese. Genetic architecture of Han Chinese is relatively similar to other East Asian populations such as JPT and KHV population from 1,000 genome. Therefore, our results are suitable for Han-Chinese populations

We have added the following sentences into the Discussion page 17 paragraph 1.

“Wei *et al* [6] showed that TWB cohort represents diverse ancestry of different province of mainland China and 99% of TWB cohort are Han Chinese. Genetic architecture of Han Chinese is relatively similar to other East Asian populations such as JPT and KHV population from 1,000 genome.”

(5) Given the emphasis on PRS and its potential interpretation/application, the introduction should include one or two sentences on the epidemiology of T2D in Taiwan--prevalence in the adult population, average age of onset, and maybe estimated effect on general health or productivity lost to this disease (e.g., DALYs or annual national cost of treatment)

Response: Thanks for the reviewers’ suggestions. We have added the following relevant information in Introduction section on page 6, paragraph 1.

“The prevalence in Taiwanese population was estimated to be 11.6% in 2016 [8] and the average age of onset for type 2 diabetes was estimated to be 59.5 years old [9]. In addition, the mean annual cost for a patient with the diabetes mellitus related major complication was estimated to be USD \$4189 [10].”

(6) In the first sentence of the introduction "Mendelian" is misspelled.

Response: The spelling error has been corrected.

Reviewer #2 (Remarks to the Author):

In the current study, Lee et al. performed a phenome-wide association study (pheWAS) for 10 diseases and 34 quantitative traits in the Taiwan Biobank (TWB). They also conducted further conditional analyses, genetic correlation and a Mendelian randomization study. Overall, they identified four novel genetic variants within or close to the HACL1, RAD21, ASH1L and GAK genes. Many of the identified loci were overlapping reflecting high pleiotropy and/or high correlation between the traits tested. It is a straightforward study with sound methodology, nevertheless, some major points both should be clarified.

Major comments

1. How did you perform GWAS for binary traits? Neither method nor supplemental methods include details about tests and models. What kind of test were performed for association? These are crucial steps; therefore, they should be included in detail.

Response: Thanks for reviewer's reminder. We conducted GWAS through logistic regression model (for binary traits) and linear regression model (for continuous traits) under the assumption of additive allelic effects of the SNP dosages via PLINK v2.0. The regression models were adjusted for age, gender and the first ten genetic principal components. The methodology for GWAS has been included in the revised manuscript in supplemental methods on page 1, paragraph 1 as followed.

“We conducted GWAS through logistic regression model (for binary traits) and linear regression model (for continuous traits) under the assumption of additive allelic effects of the SNP dosages via PLINK v2.0. The regression models were adjusted for age, gender and the first ten genetic principal components.”

2. Why did you select the first ten principal components, age and sex as covariates? Are these covariates associated with the traits? StepAIC or a similar approach can be used to select the models for each trait (<https://ashutoshtripathi.com/2019/06/10/what-is-stepaic-in-r/>).

Response: Thanks for reviewer's reminder. Covariates such as genetic principal components, age and sex are usually added to reduce confounder bias in both GWAS and PheWAS [11, 12]. For a more homogenous population like TWB, usually first ten genetic principal components are sufficient to correct for population stratification [13-15]. We did not perform any feature selection for each trait. We noticed no distinct differences between model with or without BMI and other anthropometric measurements as covariate in addition to age and sex.

3. Genome-wide significance threshold ($P \leq 5.0 \times 10^{-8}$) was defined in the European population. Would it be different in Taiwanese cohort? Please see the relevant study showing different significance level for different genetic ancestries:

Smith et al, 2022. [https://www.cell.com/ajhg/fulltext/S0002-9297\(22\)00101-X](https://www.cell.com/ajhg/fulltext/S0002-9297(22)00101-X)

Based on the new genome-wide significance level, the number of genome-wide significant loci would change for each trait.

Response: Thanks for reviewer's reminder. As TWB was sufficiently large in sample size (around 130k) when compared to UKBB (349k) and BBJ (206k), we believe that the standard genome-wide significance threshold of 5×10^{-8} is suitable. The other populations in Smith *et al* (for example: African & Native Hawaiians) had sample size of less than 30k, therefore will not have the same statistical power to detect SNPs with low frequency and small effect size.

4. Line 170; "Multi allelic SNPs and SNPs with ambiguous strands were removed from the analysis." Were these filters applied for other pheWAS analyses or specific for only PRS? In addition, filters for MAF, HWE and imputation quality were presented in PRS section. Please make sure that all filters and analyses included in methods/or supplemental method (if it is same for all analyses, you could include one general method, or separate for each analysis if the filters for specific for each analysis).

Response: Thanks for the reviewers' suggestions. We have moved the descriptions for the QC procedure for PRS in the Supplementary Method (page 3) and specified the QC procedure for PRS was conducted after the QC procedure for other PheWAS analyses (as below).

"For building PRS model: additional typical QC procedure for SNPs to build PRS models were applied. Multiallelic SNPs and SNPs with ambiguous strands were removed from the analysis. SNPs with $MAF \leq 0.01$, low imputation quality ($info < 0.3$) or deviation from HWE ($P < 10^{-6}$) were also excluded."

5. Supplemental method: What is the rationale of setting a minimum HWE p-value of 0.001 for genotyping data and HWE ($P \leq 10^{-10}$) for imputed data?

Response: Genotyping data is only used for kinship estimation, computation of principal components and genomic relation matrix, which only require smaller number of SNPs as input data, therefore we can adopt a more stringent HWE cut-off ($p = 0.001$). We believe that because imputed genetic data served as the input for all association analyses, HWE of $P \leq 10^{-10}$ is suitable and it is also a widely utilized criterion [16].

Minor comments:

1. In lines 42 and 61, please correct the spelling for Mendelian randomization. In Line 66, UKBB, one B letter is missing.

Response: We apologize for the spelling mistakes and have amended all in our revised manuscript.

2. In abstract "In total, we identified 995 significantly associated loci with more than 100 novel loci, specific to Taiwanese population. Furthermore, our analysis highlighted the genetic pleiotropy of loci associated with complex disease and associated quantitative traits. To demonstrate the versatility of our PheWAS, further extensive analysis on glyceimic phenotypes (T2D, fasting glucose and HbA1c) was performed and identified 115 significant loci"

2a. Please specify the exact number of more than 100 novel loci

2b. Did you identify 995 “variants” or “loci” in more than 100 loci?

2c. How did you figure out that these variants specific to Taiwanese population? Do not they exist in other populations? How did you determine this specificity?

Response: We appreciate the reviewer’s comment and have specified the exact number of novel variants/loci in our revised manuscript. We compared our findings with published GWAS through publicly available database, GWAS catalog. If the variants have never been reported in populations other than TWB, we will regard them as variants specific to TWB.

We have revised our abstract section on page 4, paragraph 1. We identified 995 significantly associated loci. Among them, we confirmed 860 genes or loci previously documented in the results based on the UKBB, BBJ or GWAS Catalog. At least 135 loci are novel.

3. In lines 95 and 338, genetic ancestry could be a better term or description instead of ethnicity. Ethnicity is a socially constructed concept and is used to refer to groups of people who shares a similar history and culture, such as language or history. Therefore, although it may be correlated with genetic variation, it should not be used as a proxy for genetic differences among groups.

Response: We have changed the term “ethnicity” to “ancestry” throughout the whole revised manuscript.

4. In supplemental materials, lines 22-27: how did you perform the kinship analysis and filtering? It can be clarified by including the tools and filtering thresholds for relatedness.

Response: For sample filtering, arrays with generated genotypes for < 95% of the loci were excluded. PLINK v1.9 software was used to identify samples with genetic relatedness indicating that they were from the same individual or from first-, second- or third-degree relatives. These determinations were based on evidence for cryptic relatedness from identity-by-descent status (π -hat cutoff of 0.125). We have added the above paragraph in the revised manuscript (in supplemental materials page 2):

5. Line 47, “significant loci with four novel genetic variants (HACL1, RAD21, ASH1L and GAK)”: Are the variants within these genes, or are these the closest genes to the significant hits?

Response: All the highlighted genetic variants are within the intronic regions of HACL1, RAD21, ASH1L and GAK as shown in the regional plots (supplementary Figure S10) and supplementary Table S11-S13.

Figure 5. Regional association plots, identified by comparison of Taiwan Biobank (TWB), Japan Biobank (BBJ) and UK Biobank (UKBB) genome-wide association studies (GWASs). The X-axis represents the position of loci (hg19). The Y-axis represents $-\log_{10}(P)$. Red dots are p values of variants from TWB. Green dots are p values from the BBJ. Blue dots are p values from UKBB. The red horizontal line represents $P = 5 \times 10^{-8}$. The black dashed line represents the location of the gene. A) *GCKR* gene; B) *HACL1* gene and C) *GAD2* gene.

Supplementary Figure S8. Regional plots before (left) and after (right) a conditional analysis. A) *RAD21* gene; B) *ASHIL* gene and C) *GAK* gene.

A

References

1. Karczewski, K.J., et al., *The mutational constraint spectrum quantified from variation in 141,456 humans*. *Nature*, 2020. **581**(7809): p. 434-443.
2. Jenkins, B., et al., *Peroxisomal 2-Hydroxyacyl-CoA Lyase Is Involved in Endogenous Biosynthesis of Heptadecanoic Acid*. *Molecules*, 2017. **22**(10).
3. Ishigaki, K., et al., *Large-scale genome-wide association study in a Japanese population identifies novel susceptibility loci across different diseases*. *Nat Genet*, 2020. **52**(7): p. 669-679.
4. Chen, J., et al., *The trans-ancestral genomic architecture of glycemic traits*. *Nat Genet*, 2021. **53**(6): p. 840-860.
5. Tian, D., et al., *A two-sample Mendelian randomization analysis of modifiable risk factors and intracranial aneurysms*. *Sci Rep*, 2022. **12**(1): p. 7659.
6. Lex, A., et al., *UpSet: Visualization of Intersecting Sets*. *IEEE Trans Vis Comput Graph*, 2014. **20**(12): p. 1983-92.
7. Wei, C.Y., et al., *Genetic profiles of 103,106 individuals in the Taiwan Biobank provide insights into the health and history of Han Chinese*. *NPJ Genom Med*, 2021. **6**(1): p. 10.
8. Sheen, Y.J., et al., *Trends in prevalence and incidence of diabetes mellitus from 2005 to 2014 in Taiwan*. *J Formos Med Assoc*, 2019. **118 Suppl 2**: p. S66-S73.
9. *Taiwan 2019 Almanac on Type 2 Diabetes*. Taiwanese Association of Diabetes Educators.
10. Cheng, S.W., et al., *Healthcare costs and utilization of diabetes-related complications in Taiwan: A claims database analysis*. *Medicine (Baltimore)*, 2018. **97**(31): p. e11602.

11. Uffelmann, E., et al., *Genome-wide association studies*. Nature Reviews Methods Primers, 2021. **1**(1): p. 59.
12. Pirinen, M., P. Donnelly, and C.C. Spencer, *Including known covariates can reduce power to detect genetic effects in case-control studies*. Nat Genet, 2012. **44**(8): p. 848-51.
13. Price, A.L., et al., *New approaches to population stratification in genome-wide association studies*. Nat Rev Genet, 2010. **11**(7): p. 459-63.
14. Feng, Q., et al., *A method to correct for population structure using a segregation model*. BMC Proc, 2009. **3 Suppl 7**: p. S104.
15. Kang, S.J., et al., *Assessing the impact of global versus local ancestry in association studies*. BMC Proc, 2009. **3 Suppl 7**: p. S107.
16. Mbatchou, J., et al., *Computationally efficient whole-genome regression for quantitative and binary traits*. Nat Genet, 2021. **53**(7): p. 1097-1103.

Reviewers' comments:

Reviewer #1 (Remarks to the Author):

Dr. Lee, et al, have performed extensive additional work in response to comments by both reviewers. I think the paper in its current form provides a much more comprehensive review of findings that are relevant both in terms of translational prospects for the field but just as importantly for improved representation of non-European populations in genetic studies.

I only have a couple minor comments on the revised version:

(1) Typically figures are referred to for the first time in the Results, whereas Figure 5 is mentioned first in the discussion section. Additionally, for the regional association plots in this figure, it would be helpful for the reader if the y-axis labels referenced the trait being tested (ex: "-log(P), fasting glucose") and the X-axis included the position of genes in the region (both the gene of interest any any others that fall within that locus).

(2) Regardless of whether this association & accompanying plots are mentioned earlier in the text, it would be useful to review the order and legends of all the figures--I noticed for example that the legend for the previous Figure 5 (now Figure 4) still referenced panels 5A and 5B.

Reviewer #2 (Remarks to the Author):

The authors addressed all my comments except for #3. It is mentioned that the study has enough power to detect SNPs with low frequency and small effect size. It means an even stricter significant threshold level may be needed. Bonferroni correction is used to correct for multiple testing such as one million independent variants. The concern was not about the sample size, but the independent variants tested in a diverse, non-European genetic ancestry.

3. Genome-wide significance threshold ($P \leq 5.0 \times 10^{-8}$) was defined in the European population. Would it be different in Taiwanese cohort? Please see the relevant study showing different significance level for different genetic ancestries: Smith et al, 2022. [https://www.cell.com/ajhg/fulltext/S0002-9297\(22\)00101-X](https://www.cell.com/ajhg/fulltext/S0002-9297(22)00101-X) 13 Based on the new genome-wide significance level, the number of genome-wide significant loci would change for each trait.

Response: Thanks for reviewer's reminder. As TWB was sufficiently large in sample size (around 130k) when compared to UKBB (349k) and BBJ (206k), we believe that the standard genome-wide significance threshold of 5×10^{-8} is suitable. The other populations in Smith et al (for example: African & Native Hawaiians) had sample size of less than 30k, therefore will not have the same statistical power to detect SNPs with low frequency and small effect size

Reply to Reviewers

Reviewers' comments:

Reviewer #1 (Remarks to the Author):

Dr. Lee, et al, have performed extensive additional work in response to comments by both reviewers. I think the paper in its current form provides a much more comprehensive review of findings that are relevant both in terms of translational prospects for the field but just as importantly for improved representation of non-European populations in genetic studies.

I only have a couple minor comments on the revised version:

(1) Typically, figures are referred to for the first time in the Results, whereas Figure 5 is mentioned first in the discussion section. Additionally, for the regional association plots in this figure, it would be helpful for the reader if the y-axis labels referenced the trait being tested (ex: " $-\log(P)$, fasting glucose") and the X-axis included the position of genes in the region (both the gene of interest any any others that fall within that locus).

Response: Reference on Figure 5 has been added onto the Results section page 14 paragraph 1. Figure 5 has been amended with additional labels to improve readability.

(2) Regardless of whether this association & accompanying plots are mentioned earlier in the text, it would be useful to review the order and legends of all the figures--I noticed for example that the legend for the previous Figure 5 (now Figure 4) still referenced panels 5A and 5B.

Response: Thank you for the reminder. We have reviewed the order and legends of all figures. Edit for the highlighted overlook is present on page 24.

Reviewer #2 (Remarks to the Author):

The authors addressed all my comments except for #3. It is mentioned that the study has enough power to detect SNPs with low frequency and small effect size. It means an even stricter significant threshold level may be needed. Bonferroni correction is used to correct for multiple testing such as one million independent variants. The concern was not about the sample size, but the independent variants tested in a diverse, non-European genetic ancestry.

3. Genome-wide significance threshold ($P \leq 5.0 \times 10^{-8}$) was defined in the European population. Would it be different in Taiwanese cohort? Please see the relevant study showing different significance level for different genetic ancestries: Smith et al, 2022. [https://www.cell.com/ajhg/fulltext/S0002-9297\(22\)00101-X](https://www.cell.com/ajhg/fulltext/S0002-9297(22)00101-X) 13 Based on the new genome-wide significance level, the number of genome-wide significant loci would change for each trait.

Response 1: Thanks for reviewer's reminder. As TWB was sufficiently large in sample size (around 130k) when compared to UKBB (349k) and BBJ (206k), we believe that the standard genome-wide significance threshold of 5×10^{-8} is suitable. The other populations in Smith et al (for example: African & Native Hawaiians) had sample size of less than 30k, therefore will not have the same statistical power to detect SNPs with low frequency and small effect size

Response 2: Thank you for the reviewer's comment and clarification. Indeed, as discussed in Smith *et al* 2022 [1], it is more ideal to consider the ancestry-trait specific Bonferroni-corrected significance threshold. In our study, we only consider Taiwanese population, and the maximum number of tested SNPs is 5,981,581. Therefore, the most stringent ancestry-trait specific Bonferroni-corrected significance threshold would be $0.05/5,981,581 = 8.36 \times 10^{-9}$. Among the three highlighted genes that were identified based on the traditional threshold 5×10^{-8} , only the gene *HACLI* (rs1481559294)-HbA_{1c} trait has p-value = 4.24×10^{-8} larger than the maximum ancestry-trait specific Bonferroni-corrected significance threshold. It was mentioned in the previously revised manuscript that "Jenkins *et al* showed that *Hacl1* KO mice had lower plasma and liver C17:0 fatty acid but did not observe significant difference in adipose tissue." Therefore, we believe that the gene *HACLI* is still a promising candidate gene for metabolic traits even though it missed the more stringent Bonferroni-corrected significance threshold. We have added the following discussion in the latest revised manuscript, page number 20, paragraph 2.

"As discussed in Smith *et al* [1], it is more ideal to consider the ancestry-trait specific Bonferroni-corrected significance threshold. In our study, we only consider Taiwanese population, and the maximum number of tested SNPs is 5,981,581 for all traits. Therefore, the most stringent ancestry-trait specific Bonferroni-corrected significance threshold would be 8.36×10^{-9} . Among the three highlighted genes that were identified based on the traditional threshold 5×10^{-8} only the gene *HACLI* (rs1481559294)-HbA_{1c} trait with p-value = 4.24×10^{-8} larger than the most stringent Bonferroni-corrected significance threshold in our study. However, as

mentioned above that the study of *Hac11* KO mice supports the potential involvement of *HACL1* for fatty acid, we believe that *HACL1* is still a promising candidate gene for metabolic traits,”

Reference

1. Smith, S.P., et al., *Enrichment analyses identify shared associations for 25 quantitative traits in over 600,000 individuals from seven diverse ancestries*. *Am J Hum Genet*, 2022. **109**(5): p. 871-884.